# Risk factors for severity of COVID-19 in hospital patients age 18–29 years

**Micaela Sandoval**[1,2], **Duc T. Nguyen**[1], **Farhaan S. Vahidy**[3], **Edward A. Graviss**[1,4]*

**1** Department of Pathology and Genomic Medicine, Houston Methodist Research Institute, Houston, TX, United States of America, **2** Department of Epidemiology, Human Genetics & Environmental Sciences, The University of Texas Health Science Center School of Public Health, Houston, TX, United States of America, **3** Center for Outcomes Research, Houston Methodist Research Institute, Houston, TX, United States of America, **4** Department of Surgery, Houston Methodist Hospital, Houston, TX, United States of America

* eagraviss@houstonmethodist.org

**Data Availability Statement:** Data cannot be shared publicly because of patient confidentiality concerns as imposed by the Houston Methodist Institutional Review Board. Access to de-identified data can be made to Jennifer Meeks

## Abstract

### Background

Since February 2020, over 2.5 million Texans have been diagnosed with COVID-19, and 20% are young adults at risk for SARS-CoV-2 exposure at work, academic, and social settings. This study investigated demographic and clinical risk factors for severe disease and readmission among young adults 18–29 years old, who were diagnosed at a hospital encounter in Houston, Texas, USA.

### Methods and findings

A retrospective registry-based chart review was conducted investigating demographic and clinical risk factors for severe COVID-19 among patients aged 18–29 with positive SARS-CoV-2 tests within a large metropolitan healthcare system in Houston, Texas, USA. In the cohort of 1,853 young adult patients diagnosed with COVID-19 infection at a hospital encounter, including 226 pregnant women, 1,438 (78%) scored 0 on the Charlson Comorbidity Index, and 833 (45%) were obese ($\geq$30 kg/m$^2$). Within 30 days of their diagnostic encounter, 316 (17%) patients were diagnosed with pneumonia, 148 (8%) received other severe disease diagnoses, and 268 (14%) returned to the hospital after being discharged home. In multivariable logistic regression analyses, increasing age (adjusted odds ratio [aOR] 1.1, 95% confidence interval [CI] 1.1–1.2, p<0.001), male gender (aOR 1.8, 95% CI 1.2–2.7, $p = 0.002$), Hispanic ethnicity (aOR 1.9, 95% CI 1.2–3.1, $p = 0.01$), obesity (3.1, 95% CI 1.9–5.1, p<0.001), asthma history (aOR 2.3, 95% CI 1.3–4.0, $p = 0.003$), congestive heart failure (aOR 6.0, 95% CI 1.5–25.1, $p = 0.01$), cerebrovascular disease (aOR 4.9, 95% CI 1.7–14.7, $p = 0.004$), and diabetes (aOR 3.4, 95% CI 1.9–6.2, p<0.001) were predictive of severe disease diagnoses within 30 days. Non-Hispanic Black race (aOR 1.6, 95% CI 1.0–2.4, $p = 0.04$), obesity (aOR 1.7, 95% CI 1.0–2.9, $p = 0.046$), asthma history (aOR 1.7, 95% CI 1.0–2.7, $p = 0.03$), myocardial infarction history (aOR 6.2, 95% CI 1.7–23.3, $p = 0.01$), and household exposure (aOR 1.5, 95% CI 1.1–2.2, $p = 0.02$) were predictive of 30-day readmission.

(jmeeks@houstonmethodist.org) which will be evaluated on a case by case basis in line with institutional policies.

**Funding:** This study was internally funded through the Houston Methodist Hospital, Department of Pathology and Genomic Medicine. The funders had no role in study design, data collection and analysis, decision to publish, or preparation of the manuscript.

**Competing interests:** The authors have declared that no competing interests exist.

## Conclusions

This investigation demonstrated the significant risk of severe disease and readmission among young adult populations, especially marginalized communities and people with comorbidities, including obesity, asthma, cardiovascular disease, and diabetes. Health authorities must emphasize COVID-19 awareness and prevention in young adults and continue investigating risk factors for severe disease, readmission and long-term sequalae.

## Introduction

Since February 2020, more than 27.7 million people in the United States have been diagnosed with Coronavirus Disease 2019 (COVID-19), a disease caused by the SARS-CoV-2 virus [1]. Rates of COVID-19-attributable deaths have risen disproportionately across the Southern United States, among the Hispanic population, and among adults aged 25–44 years [2]. Over 2.5 million, or 8,870 per hundred thousand population, confirmed COVID-19 cases have been reported in the state of Texas, with 20 percent coming from the greater Houston area (Harris, Fort Bend, Galveston, Waller, Montgomery, Brazoria, Liberty, and Chambers Counties) [1]. Especially in the early stages of the pandemic, the US Public Health Service (USPHS) messaging focused on risk categorization and individual risk management; adults over 65 years and patients with known co-morbidities were identified as 'high-risk' populations and prioritized in targeted health communications [3]. Young adults, meanwhile, are at increased risk of SARS-CoV-2, the causative agent of COVID-19, exposure in work, academic, and social settings. According to the Texas Department of State Health Services (DSHS), over 20% of confirmed COVID-19 cases in Texas were young adults, aged 18–29 years [4]. Specifically, the proportion of young-adult COVID-19 patients in the Houston area has markedly increased over time [5]. Among studies in young adults, only a few have incorporated longitudinal clinical data [6–8]. However, these studies were either conducted with small sample size [6, 8] or only reported preliminary data on the proportion of patients who experienced the composite event of death and mechanical ventilation [7].

The risk of SARS-CoV-2 exposure, infection, and COVID-19 disease outcomes are dependent upon both individual behaviors and societal structures. By the beginning of the second wave of the pandemic in the United States (June 2020), young adults aged 18–29 years were the least likely age group to self-report COVID-19 mitigation behaviors, including masking, handwashing, and social distancing, compared to other age groups, possibly due to lower perceived risk of severe outcomes, perceived futility of mitigation behaviors, exposure to misinformation, or peer pressure [9, 10]. An alternate hypothesis points to the risks incurred by essential workers, who are disproportionately young, low-income, minority, and immigrant, and who are often unable to dictate their workplace conditions [11], and to the infeasibility of adherence to distancing and quarantine guidelines in high-density housing, universities, and other communal settings [12].

Although young adults are at increased risk for COVID-19 exposure, the risk of immediate and long term COVID-19 outcomes and sequalae in young adults infected with SARS-CoV-2 is still largely unknown. While increased age and comorbidities have been identified as independent risk factors for hospitalization and death in the general population, the association between specific disease outcomes and other demographic and clinical risk factors among young adults diagnosed with COVID-19 have been described in only a few studies [6, 7]. The current study aimed to investigate 30-day COVID-19 disease outcomes among young adults 18–29 years old diagnosed within a large, metropolitan hospital system from March 1 to December 7, 2020.

## Methods

### Study population & setting

This study included all consecutive patients 18–29 years old diagnosed at a hospital encounter with COVID-19 between March 1 and December 7, 2020 within Houston Methodist affiliated hospitals, Houston, Texas, USA. The Houston Methodist system of hospitals consists of one central tertiary care hospital located within a large urban medical center and seven satellite hospitals. Houston Methodist received more than 120,000 admissions in 2020, and primarily serves the greater Houston area, here defined as Harris, Fort Bend, Galveston, Waller, Montgomery, Brazoria, Liberty, and Chambers counties. The greater Houston area is socioeconomically diverse, with large Hispanic, Black, and Asian populations, and more than thirty colleges and universities, and the jurisdiction encompasses urban, suburban, and rural settings. Enrollment was limited to hospital encounters, including inpatient, emergency, and observational encounters, due to availability of demographic, clinical exam and medical history data. Patients were included if they received a positive diagnostic result associated with a hospital encounter from either (1) An RNA polymerase chain reaction (PCR) test for the severe acute respiratory syndrome coronavirus 2 (SARS-CoV-2) or (2) a SARS-CoV-2 antigen test. The diagnostic encounter was defined as the hospital encounter in which a patient's first positive PCR or viral antigen respiratory sample was collected. This retrospective registry-based study was approved by the Houston Methodist institutional review board (PRO00025320) and granted a waiver of informed consent.

### Electronic medical record data collection

Demographic, geographic, and clinical data was retrieved from the Houston Methodist COVID-19 Surveillance and Outcomes Registry (CURATOR), a COVID-19-specific electronic health records (EHR) data mining and collection project. Detailed methods of the CURATOR project have been previously described [5, 13]. Demographic information, including age, patient-reported race and ethnicity, gender, parent hospital, insurance information and home address location were collected from the electronic medical record (EMR). Medical history, surgical history, body mass index (BMI), pregnancy status at encounter, hospital admission characteristics, and interventions were abstracted for initial diagnostic encounters. BMI was calculated and classified according to the United States Centers for Disease Control and Prevention (CDC) guidelines [14]. The Charlson Comorbidity Index was calculated from component medical history as a measure of overall comorbidity burden [15]. Symptom screening, exposure history, 30-day status, and diagnoses were directly abstracted from clinician notes, admissions screening, problem lists, and discharge diagnoses. Exposure history was classified as 'no known exposure', 'known exposure, non-household', and 'known household exposure', based on if the patient reported recent (within three weeks) contact with any sick person, or anyone diagnosed with COVID-19. All diagnoses were classified according to *International Classification of Diseases and Related Health Problems*, *Tenth Revision* (ICD-10-Clinical Modification [CM]) definitions. EMR data were collected by a trained abstractor and managed using the REDCap (Vanderbilt University, Nashville, TN, USA) electronic data capture tools hosted at Houston Methodist [16].

### Outcome classifications

This investigation examined the following COVID-19 outcomes: pneumonia within 30 days of initial encounter, composite disease outcomes within 30 days of initial encounter, subsequent hospital encounter within 30 days of initial discharge, and all-cause mortality. Diagnoses that

defined the composite disease outcomes were determined based on the severity described in current literature and the frequency of the disease identified in the population. The list of diagnoses was finalized in consultation with experienced clinicians. Pneumonia and component disease diagnoses were classified according to ICD-10-CM definitions, regardless of the presence of a *U07.1* code (COVID-19, virus identified). The composite disease definition included the following diagnoses: sepsis [17], myocardial infarction [18], cerebrovascular event [19], cardiac arrest, pulmonary embolism [20], thrombosis [21], acute respiratory distress syndrome (ARDS) [22], acute respiratory failure (ARF) [23], pneumothorax [24], gastrointestinal bleed [25], acute kidney injury, hypoxemia, shock, or systemic inflammatory response syndrome (SIRS) [26]. The presence of at least one of the listed diagnoses was sufficient for a 'severe disease' classification. All disease outcomes were assessed within thirty days from first COVID-19 encounter. Return to the hospital for any reason within thirty days of discharge from the initial encounter was assessed for all non-pregnant patients who were discharged home from their initial encounter; repeat hospital encounters included emergency, observational, and inpatient encounters. Patients who were discharged to another institution were excluded from this analysis (*n* = 38). Pregnant patients were excluded from 30-day repeat hospital encounters analyses, as their healthcare utilization differs significantly from the general population.

## Statistical analyses

Demographic and clinical data were reported as frequencies and proportions for categorical variables and as median and interquartile range (IQR) for continuous variables. Differences between groups were compared using the Chi-square or Fisher's exact tests for categorical variables and Kruskal-Wallis test for continuous variables. Logistic regression modeling were performed to determine the risk factors for outcomes among COVID-19 cases (composite disease outcomes, pneumonia, and returning to the hospital within 30 days of discharge); crude and adjusted odds ratios and 95% confidence intervals are provided as estimates of risk for each outcome. For all regression analyses, "missingness" was considered informative for categorical clinical variables including symptom screening and exposure history. The selection of variables for the multivariable models were conducted using the least absolute shrinkage and selection operator (LASSO) method with the cross-validation selection option and clinical importance of the covariates [27]. Briefly, all variables used in the univariable analysis were assessed by the LASSO program, which suggested good models that included the variables with the highest probability of being a risk factor. During the modeling process, the potential risk factors were discussed with senior clinicians who have extensive clinical experience in the field to ensure the biological plausibility of the selected covariates. To avoid over-fitting, some variables which were significant in the univariate analysis, but insignificant in multivariable modeling were not selected in the final model if their exclusion did not affect the diagnostic performance of the final model which was determined by a non-significant likelihood ratio test result and the area under the Receiver Operating Characteristic (ROC) curve. All analyses were performed on Stata MP version 17.0 (StataCorp LLC, College Station, TX, USA). A p-value of <0.05 was considered statistically significant.

## Geographic data collection and analyses

Cumulative COVID-19 case counts for the greater Houston area, by county and ZIP code tabulation area (ZTCA), were collected from publicly available county and local health department dashboards and the DSHS as available. Hospitalization data was collected for trauma service area Q (Austin, Colorado, Fort Bend, Harris, Matagorda, Montgomery, Walker, Waller, and Wharton Counties) from the publicly available DSHS dataset (https://dshs.texas.gov/coronavirus).

Publicly available geographic information system (GIS) datasets were collected from Texas Parks and Wildlife Department, Texas Department of Transportation, the US Census repository, and DSHS. The Area Deprivation Index, which measures relative deprivation amongst all census block groups in the state of Texas on a scale of 1–10, where one (1) is the least disadvantaged and ten (10) is the most disadvantaged [28, 29], and the Social Vulnerability Index, which measures relative vulnerability to disaster amongst all census tracts in the state of Texas [30], were utilized as location-based proxies for socioeconomic status. Heat maps were created by calculating kernel density estimates from geocoded patient-provided home addresses; low density values (<15th quantile) were truncated to preserve patient privacy. All geospatial analyses were performed on ArcGIS ArcMap version 10.7 (ESRI, Redlands, CA).

## Results

From March 1 to December 7, 2020, 22,449 patients received a positive SARS-CoV-2 PCR or viral antigen test result within the Houston Methodist system of hospitals, of whom 1,853 (8%) were both aged 18–29 years and diagnosed at a hospital encounter (Fig 1). The Houston area has experienced three distinct COVID-19 outbreak peaks, in April, July, and December 2020, which is reflected in the Houston Methodist cohort (Fig 2). Young adult Houston Methodist patients' residences were well-distributed across the greater Houston area, following the catchment areas of the seven member institutions (Fig 3). Median age of young adult patients was 24 (IQR: 21–27); the cohort was 62% women, including 226 pregnant women (12% of total population), 20% Non-Hispanic White, 32% Non-Hispanic Black, and 43% Hispanic or Latino (Table 1). This cohort was relatively healthy: 78% of patients scored 0 on the Charlson Comorbidity Index Score, though 1,252 (68%) were overweight (25–30 kg/m$^2$) or obese ($\geq$30 kg/m$^2$). The most common comorbidities were asthma (9%), mental health disorders (8%), hypertension (6%), and diabetes (5%), and 84 (5%) patients had undergone cholecystectomies. Of note, 19% of patients reported having had contact with a sick person outside of their household, and an additional 18% reported having a sick household contact.

Symptom screening results were available for 1,347 (73%) of patients; of these 550 (41%) reported no symptoms, while 797 (59%) reported at least one COVID-19 symptom, and 578 (43%) reported cough, sore throat, and/or shortness of breath (Table 2). 1,369 (74%) patients were diagnosed at an emergency department hospital encounter without an associated inpatient admission. At their diagnostic hospital encounter, 39% of patients were privately insured, 20% were Medicare or Medicaid clients, and 39% self-paid (Table 1). For the 387 patients with inpatient admissions, the median length of stay was 3 days (IQR: 2–6) (not shown). Relatively few patients received respiratory interventions (such as ventilator support) during their initial diagnostic encounter, with 11% receiving supplemental oxygen and 3% requiring intensive care. While 1,787 (96%) patients were discharged home from their initial diagnostic encounter, 263 (15%) of those returned to the hospital within thirty days of their first encounter (Fig 4). Four patients (1% of inpatient admissions) expired during their initial hospitalization, and four more expired after being discharged to another institution.

In total, 148 (8%) patients were diagnosed with at least one component disease outcome within 30 days of their first encounter, and 316 (17%) patients were diagnosed with pneumonia within 30 days of their first encounter (Table 3).

In multivariable logistic regression analysis, increasing age at encounter (continuous) was significantly associated with a composite disease outcome within thirty days of initial encounter (aOR 1.1, 95% CI 1.1–1.2, $p$<0.001) (Table 4). Pregnant women were less likely to develop composite disease outcomes (aOR 0.1, 95% CI 0.0–0.4, $p$ = 0.001), while men were more likely to develop composite disease outcomes (aOR 1.8, 95% CI 1.2–2.7, $p$ = 0.002), compared to

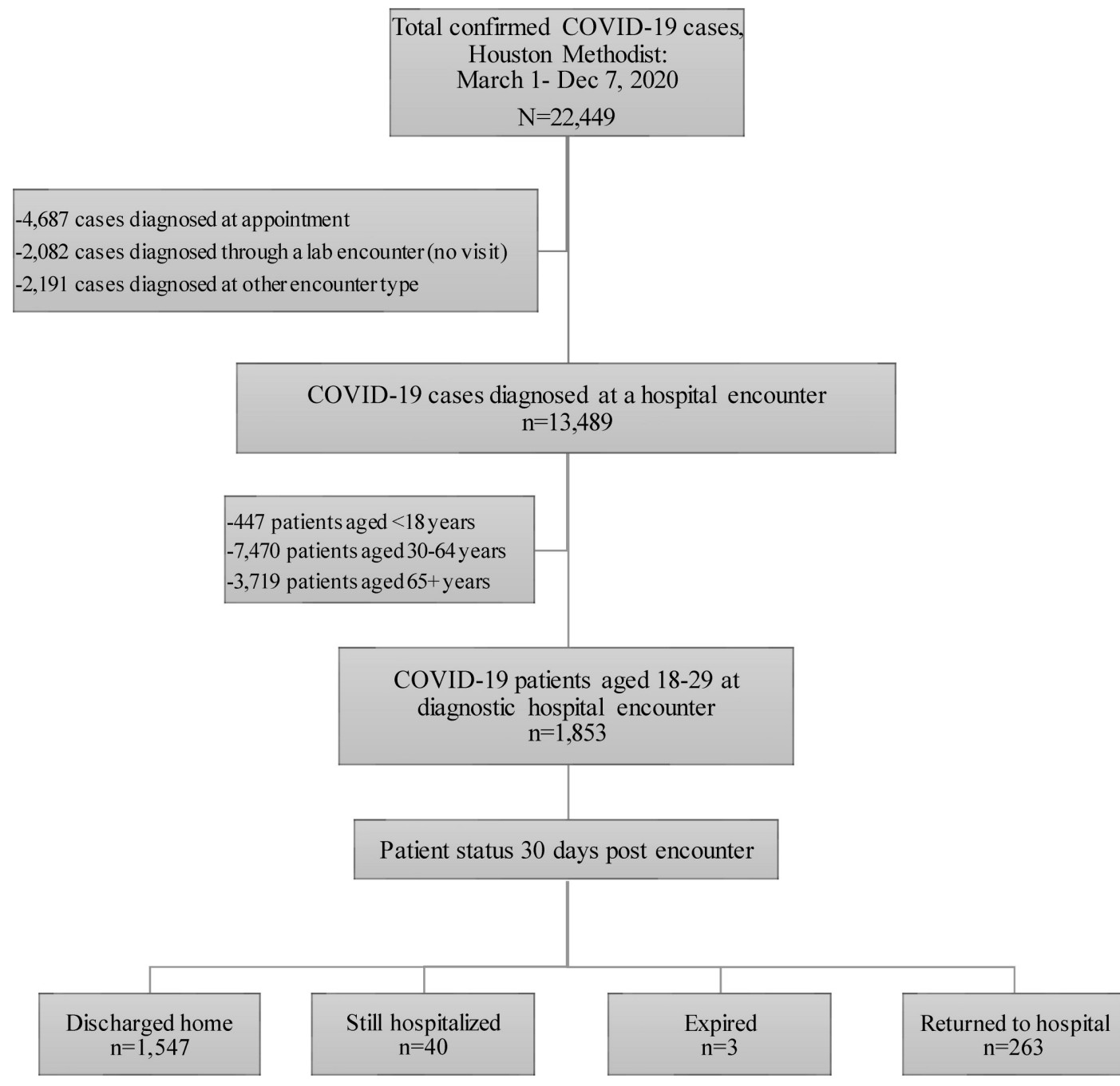

**Fig 1. Study flow.**

non-pregnant women. Only Hispanic ethnicity was positively associated with composite disease outcomes (aOR 1.9, 95% CI 1.2–3.1, $p = 0.01$), compared to non-Hispanic White patients. Admission to two of the satellite hospitals (aOR 0.4, 95% CI 0.2–0.8, $p = 0.004$ and aOR 0.3, 95% CI 0.1–0.8, $p = 0.02$) and encounters during the month of June (aOR 0.5, 95% CI 0.3–0.8, $p = 0.002$) were significantly associated with a lower odds of composite disease outcomes compared to admission to the flagship hospital and encounters during March, respectively. Missing symptom screen (aOR 1.7, 95% CI 1.1–2.6, p<0.001) and encounters during the months of

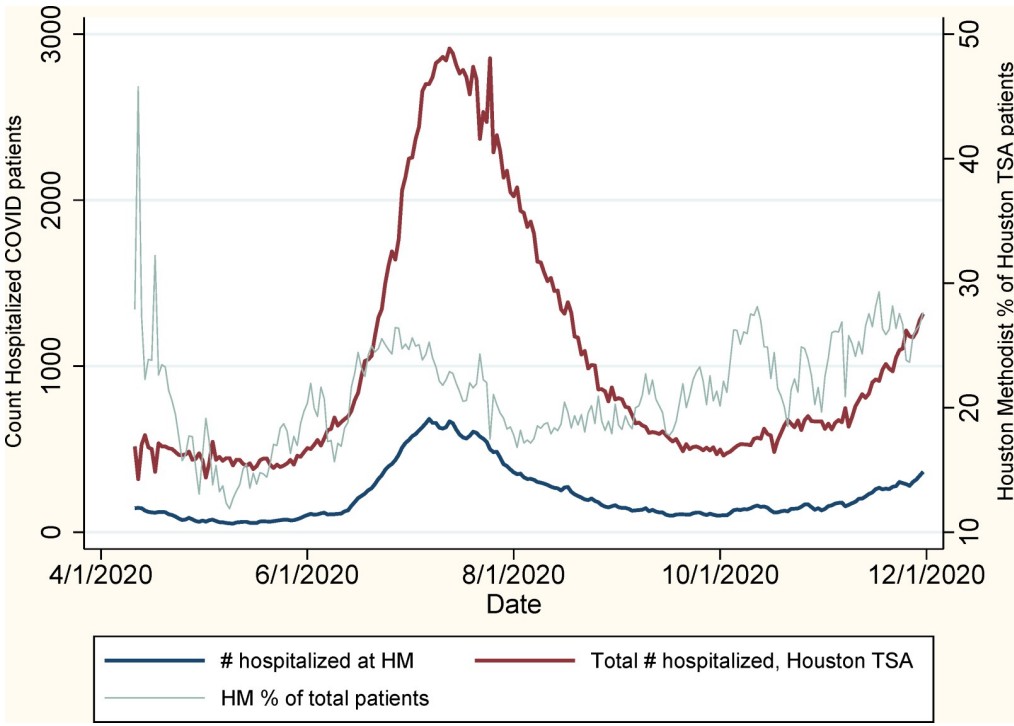

**Fig 2. Hospitalized COVID-19 patients in Houston Trauma Service Area (TSA-Q) and Houston Methodist Hospital System over time.**

September or December (aOR 2.9, 95% CI 1.4–6.1, p = 0.004 and aOR 4.9, 95% CI 1.3–18.7, p = 0.02, respectively) were significantly associated with a higher odds of composite disease outcomes compared to admission to the flagship hospital and encounters during March, respectively. Patients with Class 2 obesity (35–40 kg/m$^2$), Class 3 obesity (>40 kg/m$^2$) were at greater odds for composite disease outcomes compared to patients within the normal BMI category (aOR 3.1, 95% CI 1.9–5.1, $p<0.001$; aOR 3.8, 95% CI 2.4–6.0, $p<0.001$, respectively); asthma, myocardial infarction history, congestive heart failure, cerebrovascular disease, diabetes, and solid organ transplant history were significant risk factors for developing composite disease outcomes (Table 4).

In the multivariable logistic regression for pneumonia, patients with older age, asthma, diabetes, non-Hispanic Asian or Hispanic race/ethnicity, Class 1, Class 2, or Class 3 obesity, and a positive or missing symptom screen were associated with a higher odds of having pneumonia. Meanwhile, pregnancy, underweight, presenting to encounters during the month of June, self-pay, and reported contact with a sick person outside of their household were associated with a lower odds of being diagnosed with pneumonia (Table 5).

Among the 367 patients who were diagnosed with composite disease outcomes or pneumonia, 324 (88%) patients were diagnosed within their initial encounter, while 43 (12%) patients were diagnosed at a subsequent hospital encounter (S1 Table). Patients with a delay in composite disease or pneumonia diagnosis were more likely to be admitted at one of the satellite hospitals, admitted to the emergency department, and have non-missing symptom screens within their initial encounter. Among non-pregnant patients discharged home from their first encounter (n = 1,564), non-Hispanic Black patients were more likely to return to the hospital for any reason within thirty days of discharge from their initial encounter, compared to non-Hispanic White patients in multivariable logistic regression analysis (aOR 1.6, 95% CI 1.0–2.4,

# Houston Methodist 18-29 year old COVID-19 patients kernel density

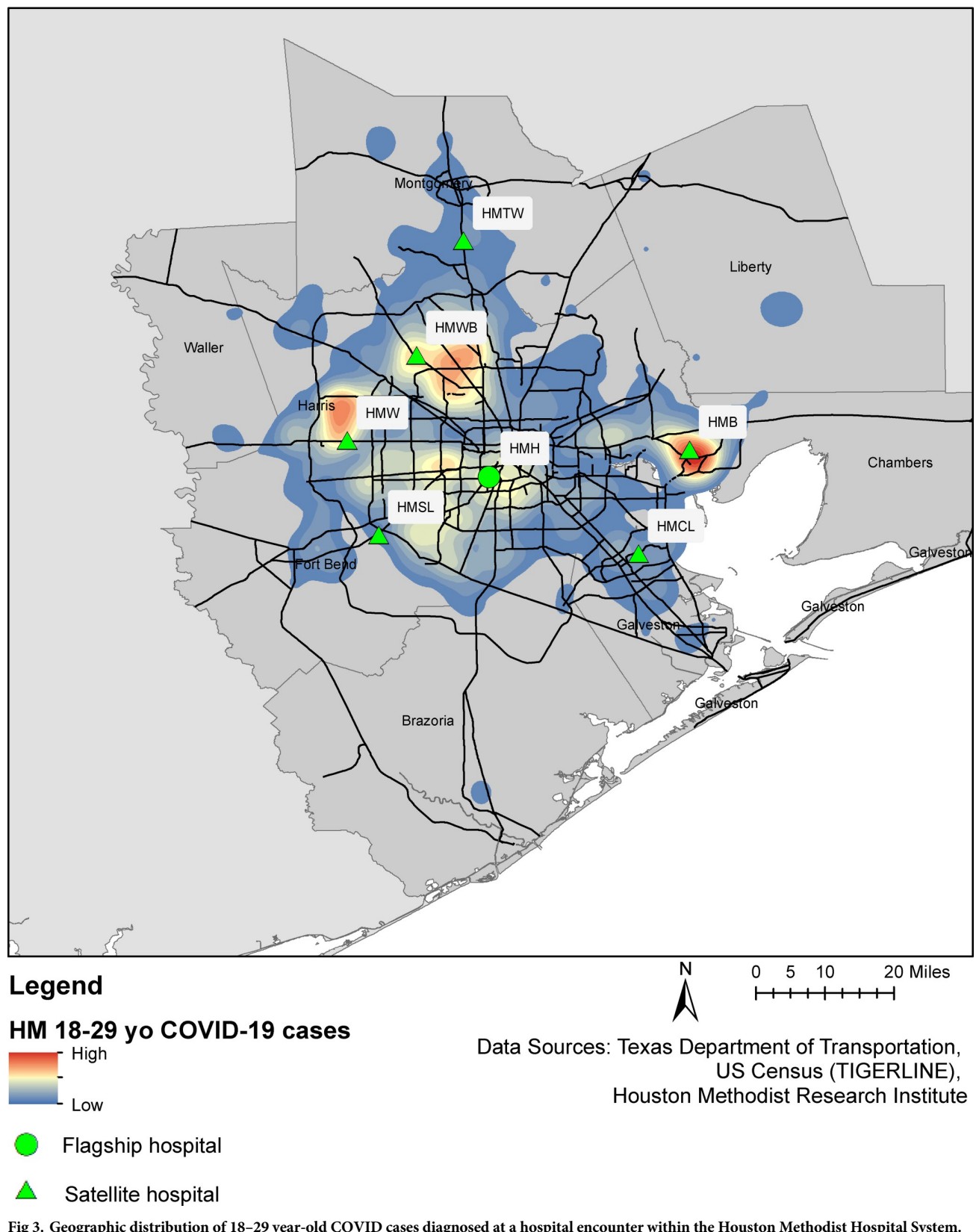

## Legend

**HM 18-29 yo COVID-19 cases**

High

Low

🟢 Flagship hospital

🔺 Satellite hospital

Data Sources: Texas Department of Transportation, US Census (TIGERLINE), Houston Methodist Research Institute

**Fig 3. Geographic distribution of 18–29 year-old COVID cases diagnosed at a hospital encounter within the Houston Methodist Hospital System.**

**Table 1. Characteristics of COVID patients aged 18–29 years, diagnosed at a hospital encounter.**

| Characteristics | Total (N = 1,853) |
|---|---|
| Age at encounter (years), median (IQR) | 24 (21–27) |
| Gender and pregnancy status | |
| Female (not pregnant) | 923 (49.8) |
| Female (pregnant) | 226 (12.2) |
| Male | 704 (38.0) |
| Race/Ethnicity | |
| NH White | 369 (19.9) |
| NH Black | 596 (32.2) |
| NH Asian | 34 (1.8) |
| NH Other Race | 17 (0.9) |
| Hispanic or Latino | 789 (42.6) |
| Unknown | 48 (2.6) |
| Area Deprivation Index (state) | |
| 1 to 2 | 395 (21.3) |
| 3 to 4 | 471 (25.4) |
| 5 to 6 | 389 (21.0) |
| 7 to 8 | 275 (14.8) |
| 9 to 10 | 287 (15.5) |
| Missing | 36 (1.9) |
| Social Vulnerability Index | |
| <20th percentile | 265 (14.3) |
| 20-39th percentile | 372 (20.1) |
| 40-59th percentile | 346 (18.7) |
| 60-79th percentile | 450 (24.3) |
| 80-99th percentile | 321 (17.3) |
| Missing | 99 (5.3) |
| Month of diagnostic encounter | |
| March | 44 (2.5) |
| April | 70 (3.8) |
| May | 58 (3.1) |
| June | 685 (37.0) |
| July | 497 (26.8) |
| August | 131 (7.1) |
| September | 83 (4.5) |
| October | 108 (5.8) |
| November | 159 (8.6) |
| December | 19 (1.0) |
| Exposure History | |
| No known exposure | 821 (44.3) |
| Known exposure, non-household | 344 (18.6) |
| Known household exposure | 335 (18.1) |
| Missing | 353 (19.1) |
| BMI (kg/m2, categorical) | |
| Underweight < = 18.5 | 34 (1.8) |
| Normal Weight 18.5–25 | 414 (22.3) |
| Overweight 25–30 | 419 (22.6) |
| Class 1 Obesity 30–35 | 363 (19.6) |

*(Continued)*

**Table 1.** (Continued)

| Characteristics | Total (N = 1,853) |
|---|---|
| Class 2 Obesity 35–40 | 227 (12.3) |
| Class 3 Obesity >40 | 243 (13.1) |
| Missing | 153 (8.3) |
| Charlson Comorbidity Index Score | |
| 0 | 1,438 (77.6) |
| 1 to 2 | 359 (19.4) |
| 3 to 4 | 23 (1.2) |
| > 4 | 33 (1.8) |
| Medical history | |
| Asthma | 166 (9.0) |
| Tuberculosis | 1 (0.1) |
| Myocardial Infarction | 14 (0.8) |
| Hypertension | 107 (5.8) |
| Congestive heart failure | 18 (1.0) |
| Cerebrovascular disease | 25 (1.3) |
| Diabetes | 98 (5.3) |
| Anemia | 81 (4.4) |
| Mental health disorders | 142 (7.7) |
| Seizure disorders | 29 (1.6) |
| Thyroid disease | 38 (2.1) |
| HIV | 13 (0.7) |
| Surgical history | |
| Cholecystectomy | 84 (4.5) |
| Appendectomy | 61 (3.3) |
| Tonsillectomy | 70 (3.8) |
| Solid organ transplant | 11 (0.6) |
| Admission Category | |
| Emergency department only | 1,368 (73.8) |
| Inpatient | 387 (20.9) |
| Observation | 66 (3.6) |
| Other | 32 (1.7) |
| Financial Class | |
| Private insurance | 727 (39.2) |
| Medicare/Medicaid | 380 (20.5) |
| Self-Pay | 727 (39.2) |
| Other | 19 (1.0) |
| Interventions | |
| Supplemental oxygen | 212 (11.4) |
| ECMO | 5 (0.3) |
| ICU stay | 49 (2.6) |
| Discharge Disposition | |
| Against Medical Advice | 19 (1.0) |
| Discharge Home | 1,787 (96.4) |
| Discharge to other hospital | 26 (1.4) |
| Discharge to SNF | 1 (0.1) |
| Discharge to LTAC | 11 (0.6) |
| Expired | 4 (0.2) |

(*Continued*)

**Table 1.** (Continued)

| Characteristics | Total (N = 1,853) |
|---|---|
| Other | 5 (0.3) |
| Therapies Administered | |
| Hydroxychloroquine | 13 (0.7) |
| Azithromycin | 181 (9.8) |
| Methylprednisolone | 47 (2.5) |
| Tocilizumab | 25 (1.3) |
| Prednisone | 25 (1.3) |
| Dexamethasone | 214 (11.5) |
| Remdesivir | 68 (3.7) |
| Any Blood Product | 50 (2.7) |
| Any IV therapy | 283 (15.3) |

Values are in number (%) for categorical variables and median (IQR) for continuous variables. IQR: Interquartile range. NH: Non-Hispanic. BMI: Body mass index. ECMO: Extracorporeal membrane oxygenation. ICU: Intensive Care Unit. SNF: Skilled nursing facility. LTAC: Long-term acute care. IV: Intravenous.

$p$ = 0.04). Admission to one of the satellite hospitals, having a sick household contact, Class 3 obesity, asthma, and history of myocardial infarction were risk factors for thirty-day return to hospital (aOR 1.9, 95% CI 1.2–3.0, $p$ = 0.004; aOR 1.5, 95% CI 1.1–2.2, $p$ = 0.02; aOR 1.7, 95% CI 1.1, 2.9, $p$ = 0.046; aOR 1.7, 95% CI 1.1–2.7, $p$ = 0.03; aOR 6.2, 95% CI1.7–23.3, $p$ = 0.01, respectively). In addition, self-payment, missing BMI, missing symptom screen, and receiving azithromycin in their initial encounter were associated with a lower likelihood of returning to the hospital Table 6).

**Table 2. Summary symptom screen results for COVID patients aged 18–29 years, diagnosed at a hospital encounter.**

| Symptoms | Total (N = 1,347*) |
|---|---|
| Any symptom | |
| Negative | 550 (40.8) |
| Positive | 797 (59.2) |
| Systemic symptoms | |
| Negative | 921 (68.4) |
| Positive | 426 (31.6) |
| Respiratory symptoms | |
| Negative | 769 (57.1) |
| Positive | 578 (42.9) |
| Neurologic symptoms | |
| Negative | 1,155 (85.7) |
| Positive | 192 (14.3) |
| Gastrointestinal symptoms | |
| Negative | 1,180 (87.6) |
| Positive | 167 (12.4) |

*Symptom screen not available in 506 patients. Values are in number (%). Systemic symptoms include fever, chills, myalgias, arthralgias, fatigue, malaise. Respiratory symptoms include cough, shortness of breath, sore throat. Neurologic symptoms include loss of smell or taste, headache. GI symptoms include nausea, vomiting, diarrhea, cramping.

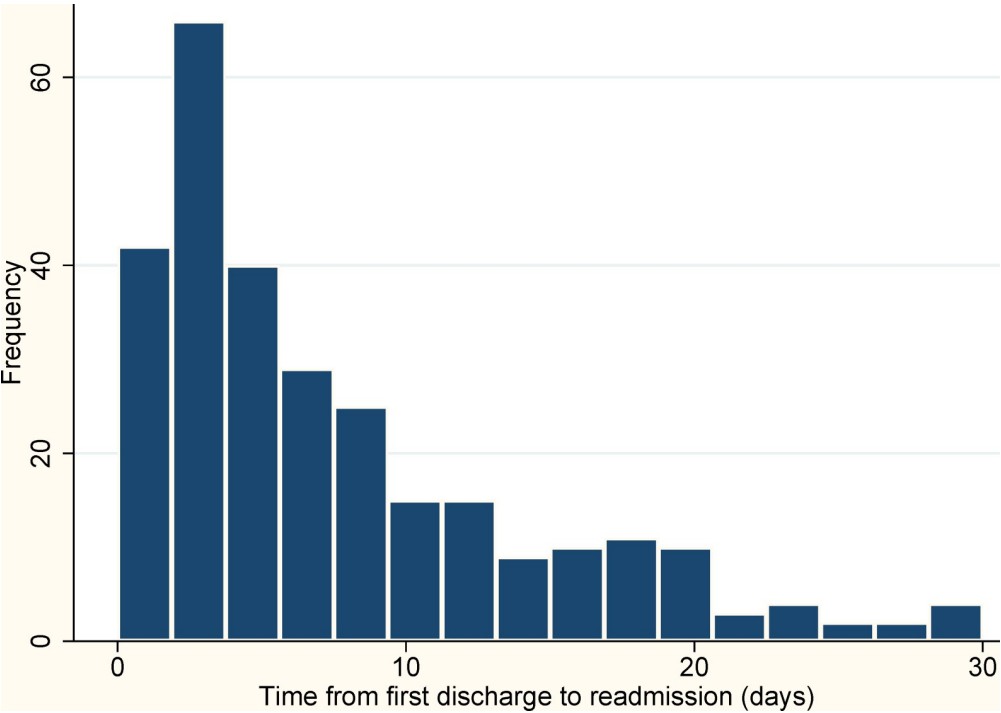

**Fig 4. Time from first discharge to subsequent hospital encounter among 18–29 year-old COVID-19 patients discharged home from initial encounter.**

**Table 3. Composite disease outcomes in COVID patients aged 18–29 years, within 30 days of initial encounters.**

| Diagnoses | Total (N = 1,853) |
|---|---|
| **Composite disease outcome** | 148 (8.0) |
| _Component disease outcomes_ | |
| Sepsis | 38 (2.1) |
| Myocardial infarction | 2 (0.1) |
| Pulmonary Embolism | 8 (0.4) |
| Cardiac Arrest | 3 (0.2) |
| Cerebrovascular event | 4 (0.2) |
| Thrombosis | 9 (0.5) |
| ARDS | 21 (1.1) |
| Pneumothorax | 3 (0.2) |
| ARF | 54 (2.9) |
| Acute kidney injury | 3 (0.2) |
| GI Bleed | 4 (0.2) |
| Hypoxemia | 86 (4.6) |
| Shock | 2 (0.1) |
| SIRS | 10 (0.5) |
| **Pneumonia** | 316 (17.1) |

Values are in number (%) for categorical variables and median (IQR) for continuous variables. ARDS = Acute Respiratory Distress Syndrome, ARF = Acute Respiratory Failure, GI = Gastrointestinal, SIRS = Systemic inflammatory response syndrome. Disease classification based on ICD-10 (CM) codes abstracted from EHR.

**Table 4. Risk factors for composite disease outcomes among COVID-19 patients 18–29 years, diagnosed at a hospital encounter.**

| Logistic regression, N = 1,853 | Univariable | | Multivariable | |
|---|---|---|---|---|
| Characteristics | OR (95% CI) | p-value | aOR (95% CI) | p-value |
| Age at encounter (years) | 1.1 (1.1–1.2) | <0.001 | 1.1 (1.1–1.2) | <0.001 |
| Gender | | | | |
| Female (not pregnant) | (ref) | | (ref) | |
| Female (pregnant) | 0.2 (0.1–0.6) | 0.003 | 0.1 (0.0–0.4) | 0.001 |
| Male | 1.6 (1.1–2.2) | 0.01 | 1.8 (1.2–2.7) | 0.002 |
| Race/Ethnicity | | | | |
| NH White | (ref) | | (ref) | |
| NH Black | 0.6 (0.4–1.1) | 0.12 | 0.6 (0.3–1.1) | 0.08 |
| NH Asian | 0.8 (0.2–3.5) | 0.76 | – | |
| NH Other Race | 3.2 (0.6–15.7) | 0.16 | – | |
| Hispanic or Latino | 1.5 (1.0–2.4) | 0.07 | 1.9 (1.2–3.1) | 0.01 |
| Unknown | 0.8 (0.2–2.9) | 0.79 | – | |
| Parent Hospital | | | | |
| Flagship hospital | (ref) | | (ref) | |
| Satellite Hospital #1 | 0.6 (0.3–1.0) | 0.046 | 0.4 (0.2–0.8) | 0.004 |
| Satellite Hospital #2 | 0.9 (0.5–1.6) | 0.73 | 1.2 (0.7–2.0) | 0.61 |
| Satellite Hospital #3 | 0.8 (0.3–1.9) | 0.55 | – | |
| Satellite Hospital #4 | 0.6 (0.3–1.3) | 0.23 | 0.3 (0.1–0.8) | 0.02 |
| Satellite Hospital #5 | 0.9 (0.5–1.4) | 0.59 | – | |
| Satellite Hospital #6 | 0.5 (0.3–0.8) | 0.01 | 0.6 (0.4–1.1) | 0.10 |
| Month of diagnostic encounter | | | | |
| March | (ref) | | (ref) | |
| April | 1.2 (0.2–7.1) | 0.81 | – | |
| May | 1.9 (0.4–10.5) | 0.44 | – | |
| June | 1.1 (0.3–4.8) | 0.89 | 0.5 (0.3–0.8) | 0.002 |
| July | 2.0 (0.5–8.7) | 0.34 | – | |
| August | 2.5 (0.5–11.3) | 0.25 | – | |
| September | 3.1 (0.7–14.8) | 0.15 | 2.9 (1.4–6.1) | 0.004 |
| October | 2.3 (0.5–11.0) | 0.29 | – | |
| November | 2.5 (0.5–11.1) | 0.24 | – | |
| December | 5.5 (0.9–33.0) | 0.06 | 4.9 (1.3–18.7) | 0.02 |
| Social Vulnerability Index | | | | |
| <20th percentile | (ref) | | (ref) | |
| 20-39th percentile | 0.8 (0.4–1.4) | 0.41 | – | – |
| 40-59th percentile | 0.8 (0.4–1.4) | 0.41 | 0.7 (0.4–1.1) | 0.14 |
| 60-79th percentile | 0.8 (0.5–1.4) | 0.41 | – | – |
| 80-99th percentile | 0.9 (0.5–1.6) | 0.68 | – | – |
| Missing | 1.5 (0.7–3.1) | 0.26 | – | – |
| Financial Class | | | | |
| Private insurance | (ref) | | (ref) | |
| Medicare/Medicaid | 0.8 (0.5–1.2) | 0.26 | – | – |
| Self-Pay | 0.8 (0.6–1.2) | 0.25 | – | – |
| Other | 0.6 (0.1–4.2) | 0.57 | – | – |
| BMI (kg/m$^2$), categorical | | | | |
| Underweight ≤18.5 | 1.4 (0.3–6.2) | 0.68 | – | – |
| Normal Weight 18.5–25 | (ref) | | (ref) | |

*(Continued)*

**Table 4.** (Continued)

| Logistic regression, N = 1,853 | Univariable | | Multivariable | |
|---|---|---|---|---|
| Characteristics | OR (95% CI) | p-value | aOR (95% CI) | p-value |
| Overweight 25–30 | 1.5 (0.8–2.7) | 0.23 | – | – |
| Class 1 Obesity 30–35 | 1.8 (1.0–3.3) | 0.07 | – | – |
| Class 2 Obesity 35–40 | 3.5 (1.9–6.4) | <0.001 | 3.1 (1.9–5.1) | <0.001 |
| Class 3 Obesity >40 | 4.1 (2.3–7.3) | <0.001 | 3.8 (2.4–6.0) | <0.001 |
| Missing | 0.9 (0.3–2.3) | 0.82 | – | – |
| Medical History | | | | |
| Asthma | 1.9 (1.2–3.1) | 0.01 | 2.3 (1.3–4.0) | 0.003 |
| Myocardial Infarction | 16.2 (5.5–47.2) | <0.001 | 5.8 (1.2–27.7) | 0.03 |
| Hypertension | 1.8 (1.0–3.3) | 0.048 | 0.6 (0.3–1.2) | 0.15 |
| Congestive heart failure | 9.7 (3.8–24.9) | <0.001 | 6.0 (1.5–25.1) | 0.01 |
| Cerebrovascular disease | 5.7 (2.4–13.4) | <0.001 | 4.9 (1.7–14.7) | 0.004 |
| Diabetes | 3.7 (2.3–6.2) | <0.001 | 3.4 (1.9–6.2) | <0.001 |
| Anemia | 1.3 (0.6–2.7) | 0.52 | – | – |
| Mental health disorders | 1.1 (0.6–2.0) | 0.83 | – | – |
| Seizure disorders | 1.3 (0.4–4.5) | 0.64 | – | – |
| Thyroid disease | 2.2 (0.9–5.4) | 0.08 | 1.5 (0.5–4.1) | 0.45 |
| HIV | 1.9 (0.4–8.7) | 0.30 | – | – |
| Surgical History | | | | |
| Cholecystectomy | 1.0 (0.5–2.3) | 0.91 | – | – |
| Appendectomy | 1.8 (0.8–3.8) | 0.14 | – | – |
| Tonsillectomy | 0.3 (0.1–1.4) | 0.13 | – | – |
| Solid organ transplant | 6.7 (1.9–23.3) | 0.003 | 5.6 (1.2–26.8) | 0.03 |
| Exposure History | | | | |
| No known exposure | (ref) | | (ref) | |
| Known exposure, non-household | 0.8 (0.5–1.4) | 0.47 | – | – |
| Known HH exposure | 1.5 (1.0–2.4) | 0.08 | 1.1 (0.7–1.7) | 0.70 |
| Missing | 2.3 (1.5–3.4) | <0.001 | – | |
| Symptom Screen | | | | |
| Negative | (ref) | | (ref) | |
| Positive | 1.1 (0.7–1.6) | 0.80 | – | – |
| Missing | 1.5 (1.0–2.4) | 0.052 | 1.7 (1.1–2.6) | <0.001 |

OR: *Odds Ratio*. aOR: *Adjusted Odds Ratio*. CI: *Confidence Interval*. NH: *Non-Hispanic*. BMI: *Body mass index*. HIV: *Human Immunodeficiency Virus*. HH: *Household*. Model notes: include all 18–29 year-old individuals diagnosed with COVID-19 at a hospital encounter. Outcome: Severe disease diagnosed within 30 days of first encounter (not including pneumonia). Multivariable model: includes data from records with complete data sets for all included variables; aORs generated from multivariable models; *C-statistic*: 0.82.

## Discussion

In this cohort of 18–29 year-old COVID-19 patients diagnosed at a hospital encounter, nearly 20% experienced composite disease outcomes, including pneumonia, within 30 days of their initial visit (n = 366). While all patients were PCR positive and potentially infectious at some point during their diagnostic encounter, only 43% (797/1,853) reported COVID-19 symptoms at admission. The high proportion of patients with emergency encounter types (74%) and patients reporting recent exposure to a sick contact (779/1853 = 36.7%) coupled with a low proportion of patients with severe symptoms at admission could indicate community utilization of the emergency department as a primary source for COVID-19 screening and

**Table 5. Risk factors for pneumonia among COVID-19 patients 18–29 years, diagnosed at a hospital encounter.**

| Logistic regression, N = 1,853 | Univariable | | Multivariable | |
|---|---|---|---|---|
| Characteristics | OR (95% CI) | p-value | aOR (95% CI) | p-value |
| Age at encounter (years) | 1.2 (1.1–1.2) | <0.001 | 1.1 (1.1–1.2) | <0.001 |
| Gender | | | | |
| Female (not pregnant) | (ref) | | (ref) | |
| Female (pregnant) | 0.3 (0.1–0.5) | <0.001 | 0.1 (0.1–0.2) | <0.001 |
| Male | 1.2 (0.9–1.5) | 0.27 | 1.2 (0.9–1.6) | 0.12 |
| Race/Ethnicity | | | | |
| NH White | (ref) | | (ref) | |
| NH Black | 1.1 (0.7–1.6) | 0.76 | 0.9 (0.6–1.3) | 0.43 |
| NH Asian | 4.0 (1.9–8.7) | <0.001 | 4.7 (2.0–11.0) | <0.001 |
| NH Other Race | 3.2 (0.8–12.7) | 0.10 | – | – |
| Hispanic or Latino | 2.0 (1.4–2.9) | <0.001 | 1.8 (1.3–2.5) | 0.002 |
| Unknown | 1.9 (0.9–4.2) | 0.09 | – | |
| Parent Hospital | | | | |
| Flagship hospital | (ref) | | (ref) | |
| Satellite Hospital #1 | 0.9 (0.6–1.3) | 0.50 | – | – |
| Satellite Hospital #2 | 1.0 (0.7–1.5) | 0.92 | – | – |
| Satellite Hospital #3 | – | | – | – |
| Satellite Hospital #4 | 0.8 (0.5–1.4) | 0.43 | 0.7 (0.4–1.2) | 0.17 |
| Satellite Hospital #5 | 0.7 (0.5–1.1) | 0.11 | – | – |
| Satellite Hospital #6 | 0.8 (0.6–1.2) | 0.29 | – | – |
| Month of diagnostic encounter | | | | |
| March | (ref) | | (ref) | |
| April | 1.3 (0.5–3.5) | 0.62 | – | – |
| May | 1.8 (0.7–4.9) | 0.25 | – | – |
| June | 0.9 (0.4–2.1) | 0.78 | 0.7 (0.6–0.9) | 0.04 |
| July | 1.2 (0.5–2.8) | 0.65 | – | – |
| August | 0.8 (0.3–2.0) | 0.59 | – | – |
| September | 1.0 (0.4–2.6) | 0.93 | – | – |
| October | 1.5 (0.6–3.7) | 0.42 | – | – |
| November | 1.0 (0.4–2.4) | 0.93 | 0.6 (0.4–1.0) | 0.08 |
| December | 1.8 (0.5–6.8) | 0.36 | 2.3 (0.7–7.4) | 0.189 |
| Social Vulnerability Index | | | | |
| <20th percentile | (ref) | | (ref) | |
| 20-39th percentile | 0.9 (0.6–1.4) | 0.59 | – | – |
| 40-59th percentile | 1.1 (0.7–1.7) | 0.67 | – | – |
| 60-79th percentile | 0.9 (0.6–1.3) | 0.55 | – | – |
| 80-99th percentile | 1.0 (0.7–1.6) | 0.92 | – | – |
| Missing | 0.7 (0.4–1.4) | 0.29 | – | – |
| Financial Class | | | | |
| Private insurance | (ref) | | (ref) | |
| Medicare/Medicaid | 0.6 (0.4–0.9) | 0.01 | – | – |
| Self-Pay | 0.9 (0.7–1.1) | 0.28 | 0.6 (0.5–0.8) | 0.002 |
| Other | 0.5 (0.1–2.2) | 0.35 | – | |
| BMI (kg/m$^2$), categorical | | | | |
| Underweight ≤18.5 | 0.4 (0.1–3.0) | 0.38 | 0.5 (0.3–0.9) | 0.015 |
| Normal Weight 18.5–25 | (ref) | | (ref) | |

*(Continued)*

**Table 5.** (Continued)

| Logistic regression, N = 1,853 | Univariable | | Multivariable | |
|---|---|---|---|---|
| Characteristics | OR (95% CI) | p-value | aOR (95% CI) | p-value |
| Overweight 25–30 | 1.9 (1.2–3.1) | 0.01 | – | – |
| Class 1 Obesity 30–35 | 3.3 (2.1–5.3) | <0.001 | 1.7 (1.2–2.6) | 0.01 |
| Class 2 Obesity 35–40 | 3.4 (2.1–5.5) | <0.001 | 1.7 (1.1–2.8) | 0.02 |
| Class 3 Obesity >40 | 7.3 (1.7–5.2) | <0.001 | 4.0 (2.7–6.1) | <0.001 |
| Missing | 3.0 (1.7–5.2) | <0.001 | 1.7 (1.0–2.8) | 0.07 |
| Medical History | | | | |
| Asthma | 1.4 (1.0–2.1) | 0.06 | 2.4 (1.1–5.1) | 0.03 |
| Myocardial Infarction | 1.3 (0.4–4.8) | 0.67 | – | – |
| Hypertension | 2.2 (1.4–3.4) | <0.001 | 1.5 (0.9–2.4) | 0.14 |
| Congestive heart failure | 1.4 (0.5–4.3) | 0.56 | – | – |
| Cerebrovascular disease | 1.2 (0.5–3.3) | 0.70 | – | – |
| Diabetes | 2.8 (1.8–4.3) | <0.001 | 2.1 (1.3–3.5) | 0.003 |
| Anemia | 0.6 (0.3–1.2) | 0.15 | – | – |
| Mental health disorders | 1.0 (0.7–1.6) | 0.86 | – | – |
| Seizure disorders | 0.8 (0.3–2.2) | 0.64 | – | – |
| Thyroid disease | 1.8 (0.8–3.7) | 0.13 | – | – |
| HIV | 2.7 (0.9–8.2) | 0.07 | – | – |
| Surgical History | | | | |
| Cholecystectomy | 1.6 (0.9–2.6) | 0.09 | – | – |
| Appendectomy | 1.1 (0.6–2.1) | 0.84 | – | – |
| Tonsillectomy | 1.1 (0.6–2.1) | 0.73 | – | – |
| Solid organ transplant | 0.5 (0.1–3.8) | 0.49 | – | – |
| Exposure History | | | | |
| No known exposure | (ref) | | (ref) | |
| Known exposure, non-household | 0.7 (0.5–1.1) | 0.12 | 0.6 (0.4–0.9) | 0.02 |
| Known HH exposure | 1.7 (1.2–2.3) | 0.00 | 1.2 (0.9–1.7) | 0.22 |
| Missing | 1.3 (0.96–1.8) | 0.09 | – | – |
| Symptom Screen | | | | |
| Negative | (ref) | | (ref) | |
| Positive | 1.8 (1.3–2.4) | <0.001 | 1.5 (1.0–2.1) | 0.04 |
| Missing | 2.0 (1.4–2.8) | <0.001 | 2.0 (1.3–2.9) | 0.001 |

OR: *Odds Ratio*. aOR *Adjusted Odds Ratio*. CI: *Confidence Interval*. NH: *Non-Hispanic*. BMI: *Body mass index*. HIV: *Human Immunodeficiency Virus*. HH: *Household*.
Model notes: include all 18–29 year-old individuals diagnosed with COVID-19 at a hospital encounter. Outcome: Pneumonia diagnosed within 30 days of first encounter. Multivariable model: includes data from records with complete data sets for all included variables; aORs generated from multivariable models; *C-statistic*: 0.77.

diagnostics, especially among uninsured or underinsured populations. Of note, pregnant women were far less likely to be diagnosed with pneumonia or other disease indicators than either non-pregnant women or men, possibly due to being regularly screened for COVID-19 during their routine prenatal or labor visits which are unrelated to COVID-19. Pregnant patients may therefore represent a population of largely subclinical COVID-19 cases who were diagnosed incidentally, and future studies are needed to investigate long-term maternal and fetal outcomes of symptomatic and asymptomatic patients.

Another startling finding was that 14% of young adult patients discharged home after being diagnosed with COVID-19 returned to the hospital for additional reasons within thirty days.

**Table 6. Risk factors for subsequent hospital encounter within 30 days among non-pregnant COVID-19 patients 18–29 years, diagnosed at a hospital encounter.**

| Logistic regression, N = 1,564 | Univariable | | Multivariable | |
|---|---|---|---|---|
| Characteristics | OR (95% CI) | p-value | aOR (95% CI) | p-value |
| Age at encounter (years) | 1.0 (1.0–1.1) | 0.20 | 1.0 (1.0–1.1) | 0.15 |
| Gender | | | | |
| Female (not pregnant) | (ref) | | (ref) | |
| Male | 0.9 (0.7–1.3) | 0.64 | – | – |
| Race/Ethnicity | | | | |
| NH White | (ref) | | (ref) | |
| NH Black | 1.3 (0.8–1.9) | 0.27 | 1.6 (1.0–2.4) | 0.04 |
| NH Asian | 0.8 (0.2–2.8) | 0.73 | – | – |
| NH Other Race | 0.8 (0.1–6.3) | 0.81 | – | – |
| Hispanic or Latino | 1.0 (0.7–1.6) | 0.82 | 1.4 (0.9–2.1) | 0.09 |
| Unknown | 0.4 (0.1–1.6) | 0.18 | – | |
| Parent Hospital | | | | |
| Flagship hospital | (ref) | | (ref) | |
| Satellite Hospital #1 | 1.3 (0.8–2.0) | 0.24 | 1.9 (1.2–3.0) | 0.004 |
| Satellite Hospital #2 | 0.9 (0.5–1.4) | 0.58 | – | |
| Satellite Hospital #3 | 1.5 (0.8–3.0) | 0.24 | – | |
| Satellite Hospital #4 | 1.1 (0.6–1.9) | 0.85 | 1.7 (0.9–3.3) | 0.10 |
| Satellite Hospital #5 | 1.0 (0.6–1.6) | 0.92 | – | |
| Satellite Hospital #6 | 0.8 (0.5–1.3) | 0.48 | 0.7 (0.5–1.1) | 0.13 |
| Month of diagnostic encounter | | | | |
| March | (ref) | | (ref) | |
| April | 0.6 (0.2–2.2) | 0.42 | – | – |
| May | 1.5 (0.4–5.1) | 0.50 | – | – |
| June | 1.2 (0.4–3.1) | 0.75 | – | – |
| July | 1.0 (0.4–2.6) | 0.95 | 0.8 (0.5–1.1) | 0.15 |
| August | 0.8 (0.3–2.6) | 0.75 | 0.6 (0.3–1.2) | 0.12 |
| September | 1.4 (0.4–4.5) | 0.56 | – | – |
| October | 1.1 (0.4–3.5) | 0.83 | – | – |
| November | 0.7 (0.2–2.1) | 0.53 | 0.6 (0.3–1.2) | 0.17 |
| December | 3.1 (0.8–12.7) | 0.12 | – | |
| Social Vulnerability Index | | | | |
| <20th percentile | (ref) | | (ref) | |
| 20-39th percentile | 1.8 (1.1–3.0) | 0.02 | 1.4 (0.97–2.0) | 0.07 |
| 40-59th percentile | 1.3 (0.7–2.2) | 0.42 | – | – |
| 60-79th percentile | 1.2 (0.7–2.0) | 0.51 | – | – |
| 80-99th percentile | 1.2 (0.7–2.1) | 0.54 | – | – |
| Missing | 1.4 (0.7–2.9) | 0.38 | – | – |
| Financial Class | | | | |
| Private insurance | (ref) | | (ref) | |
| Medicare/Medicaid | 1.5 (1.0–2.2) | 0.06 | – | |
| Self-Pay | 0.7 (0.5–1.0) | 0.08 | 0.6 (0.5–0.9) | 0.01 |
| Other | 0.9 (0.2–3.9) | 0.86 | – | |
| BMI (kg/m$^2$), categorical | | | | |
| Underweight $\leq$18.5 | 1.3 (0.4–3.9) | 0.65 | – | – |
| Normal Weight 18.5–25 | (ref) | | (ref) | |
| Overweight 25–30 | 1.3 (0.9–2.1) | 0.17 | 1.5 (1.0–2.4) | 0.06 |

*(Continued)*

**Table 6.** (Continued)

| Logistic regression, N = 1,564 | Univariable | | Multivariable | |
|---|---|---|---|---|
| Characteristics | OR (95% CI) | p-value | aOR (95% CI) | p-value |
| Class 1 Obesity 30–35 | 1.3 (0.8–2.0) | 0.30 | 1.4 (0.9–2.3) | 0.14 |
| Class 2 Obesity 35–40 | 1.4 (0.8–2.3) | 0.24 | 1.5 (0.9–2.6) | 0.13 |
| Class 3 Obesity >40 | 1.5 (0.9–2.4) | 0.13 | 1.7 (1.1–2.9) | 0.046 |
| Missing | 0.3 (0.1–0.7) | 0.01 | 0.3 (0.1–0.9) | 0.02 |
| Medical History | | | | |
| Asthma | 1.6 (1.0–2.5) | 0.04 | 1.7 (1.0–2.7) | 0.03 |
| Myocardial Infarction | 5.5 (1.7–18.1) | 0.01 | 6.2 (1.7–23.3) | 0.01 |
| Hypertension | 1.8 (1.0–3.0) | 0.04 | – | – |
| Congestive heart failure | 3.6 (1.2–11.0) | 0.02 | – | – |
| Cerebrovascular disease | 1.9 (0.6–5.7) | 0.28 | – | – |
| Diabetes | 1.0 (0.5–2.0) | 0.93 | – | – |
| Anemia | 2.1 (1.1–3.9) | 0.20 | – | – |
| Mental health disorders | 1.9 (1.2–3.0) | 0.01 | 1.7 (0.99–2.9) | 0.051 |
| Seizure disorders | 3.5 (1.5–8.4) | 0.004 | 2.6 (0.97–6.9) | 0.06 |
| Thyroid disease | 1.6 (0.7–4.0) | 0.29 | – | – |
| HIV | 3.3 (1.0–10.9) | 0.06 | 4.2 (0.9–18.8) | 0.06 |
| Surgical History | | | | |
| Cholecystectomy | 1.3 (0.7–2.4) | 0.44 | – | – |
| Appendectomy | 0.8 (0.3–2.0) | 0.61 | – | – |
| Tonsillectomy | 0.9 (0.4–2.0) | 0.82 | – | – |
| Solid organ transplant | 1.6 (0.3–7.7) | 0.55 | – | – |
| Exposure History | | | | |
| No known exposure | (ref) | | (ref) | |
| Known exposure, non-household | 1.0 (0.7–1.4) | 0.84 | – | – |
| Known HH exposure | 1.3 (0.9–1.9) | 0.14 | 1.5 (1.1–2.2) | 0.02 |
| Missing | 0.6 (0.4–1.0) | 0.04 | – | |
| Symptom Screen | | | | |
| Negative | (ref) | | (ref) | |
| Positive | 1.0 (0.7–1.4) | 0.98 | – | – |
| Missing | 0.4 (0.3–0.7) | <0.001 | 0.4 (0.2–0.6) | <0.001 |
| Admission Category (initial encounter) | | | | |
| Emergency department only | (ref) | | (ref) | |
| Inpatient | 0.6 (0.3–0.9) | 0.01 | – | – |
| Observation | 0.9 (0.4–2.1) | 0.79 | – | – |
| Therapy administered at initial encounter | | | | |
| Supplemental oxygen | 0.5 (0.3–0.9) | 0.03 | – | |
| Azithromycin | 0.3 (0.1–0.7) | 0.003 | 0.3 (0.1–0.6) | 0.001 |
| Dexamethasone | 0.8 (0.5–1.3) | 0.32 | – | – |
| Remdesivir | 0.3 (0.1–1.1) | 0.06 | – | – |
| Any IV therapy | 0.6 (0.4–1.0) | 0.07 | – | – |

OR: *Odds Ratio*. aOR *Adjusted Odds Ratio*. CI: *Confidence Interval*. NH: *Non-Hispanic*. BMI: *Body mass index*. HIV: *Human Immunodeficiency Virus*. HH: *Household*. Model notes: include all non-pregnant 18–29 year-olds diagnosed with COVID-19 at a hospital encounter discharged home after initial encounter. Outcome: Patient readmitted within 30 days of discharge from initial encounter. Time to subsequent hospital encounter defined as days from first discharge (discharge date for diagnostic hospital encounter) to next hospital encounter. Multivariable model: includes data from records with complete data sets for all included variables; aORs generated from multivariable models; *C-statistic*: 0.72.

Most (52%) of the non-pregnant patients who returned within 30 days, did so in the first five days following their initial discharge (120/229). While relatively few patients received drugs such as dexamethasone or remdesivir, azithromycin treatment in the initial encounter was a risk factor for returning to the hospital. Overall, 12% of patients diagnosed with composite disease outcomes within thirty days were diagnosed at a subsequent encounter. Notably, non-Hispanic Black patients were not more likely to be diagnosed with composite disease outcomes or pneumonia, but they were likely to return to the hospital within thirty days, compared to White patients; in contrast, Hispanic patients were more likely to be diagnosed with composite disease outcomes or pneumonia compared to White patients. Among patients with composite disease outcomes, neither Black race nor Hispanic ethnicity were associated with delayed diagnosis. Considering the disproportionate COVID-19-related mortality shouldered by the Black and Hispanic communities in the Southern United States,[2] these results could point to under-diagnosis of severe disease in Black patients, increased admissions for non-COVID-19-related problems among Black patients, or even increased anxiety among minority patients receiving a positive test. These associations between race and ethnicity and poor health outcomes are unlikely to be based on biological vulnerability, and additional epidemiologic and behavioral research will be needed to understand the intersection between race and ethnicity, socioeconomic factors, barriers to accessing healthcare, and COVID-19 disease risk. Of note, our analysis found that patients with a missing BMI had lower odds of readmission compared with patients having a normal BMI. In fact, the small group of missing BMI measurements included patients having much fewer underlying conditions with most of the patients having CCI $\leq 2$. Given the small sample size of the missing BMI category, we could not rule out the possibility that those patients actually had normal BMIs and the significance seen in the readmission difference with the current normal BMI group occurred by chance.

Among these relatively healthy young adults, obesity, asthma, diabetes, and cardiovascular disease emerged as important risk factors for both pneumonia and other disease indicators. This finding is especially concerning due to the intersectionality of race and ethnicity, economic status, and environmental factors contributing to higher prevalence of these conditions among members of marginalized communities [31–33]. The synergistic internal and external social determinants of health such as stress, low socioeconomic status, access to quality care, and trust in healthcare providers that drive increased prevalence of diabetes, asthma, and heart disease among Black and Hispanic communities may also increase the incidence of poor COVID-19 outcomes in these populations, and the presence of any of these chronic conditions worsens the COVID-19 disease course in the individual. The location-based relative vulnerability index was not a significant factor in any of the outcomes assessed. This observation could be a product of the demographics of this large not-for-profit hospital system's catchment population, or it could be that neither the CDC's Social Vulnerability Index nor the Area Deprivation Index are appropriate proxies for sociodemographic vulnerability during this type of public health crisis in our population. Population-based research is needed to assess the effects of geography and socioeconomic status on the risk of COVID-19 infection, advancement to severe disease, and poor health outcomes.

Although the adverse outcomes in COVID-19 patients may also be affected by certain hospital-related factors (such as catchment population, staff experience, referral or community hospital, and equipment capacity), published data on these issues appear to be unavailable. Therefore, we attempted to address the issue by evaluating the hospitalization to a satellite hospital versus the flagship hospital is relevant and warrants continued investigation. Our findings of having increased number of patients with composite disease outcomes in patients hospitalized presenting to our flagship hospital is consistent with the fact that our flagship hospital is a tertiary hospital located in a large medical center and received more severe referrals needing

higher levels of care than satellites hospitals, especially early in the outbreak. Additionally, the flagship hospital is centrally located and serves high population-density, urban communities surrounding the medical center. Meanwhile, given the higher odds of subsequent hospital encounters within 30 days among patients who initially presented to the in the satellite hospitals compared to the flagship hospital, we could not rule out the contribution of the staff experience level and equipment capacity. Further studies on this issue would be programmatically appropriate.

This study has several important strengths. First, our study examined a large, diverse, and well-characterized population of young adults with COVID-19 in a major United Stated metropolitan area. Additionally, we were able to assess thirty-day patient statuses, including subsequent hospital encounters, and to capture a range of disease outcomes. The previously published research articles describing risk factors for severe COVID-19 in young adults included a combined total of fewer than 1,500 patients, and primarily included information from the diagnostic encounter [6–8]. Our findings constitute a substantial addition to the existing knowledge base because we not only included data for young adults diagnosed at both inpatient and emergency department encounters, but also collected longitudinal outcome data, thus allowing us to characterize patients at all stages of disease progression. The study also has several limitations. Because the entry point was a positive SARS-CoV-2 test within a hospital encounter, we were not able to assess risk factors for COVID-19 infection, and our results may not be generalizable for all young adults with COVID-19. Since Houston Methodist is a not-for-profit hospital system, this cohort may represent fewer uninsured or underinsured patients than the general population of COVID-19 cases. Furthermore, because state and local health departments' publicly available case data lacks reliable demographic information, such as race and ethnicity, we were not able to determine if our cohort was representative of COVID-19 patients in the greater Houston Area. Finally, our findings may underestimate the actual 30-day outcomes because we cannot rule out the possibility that, following their initial encounter, patients sought further care at an institution outside of the Houston Methodist Hospital System, where the outcome data was not available to the research team. Given the distinctions between the diagnoses that are related versus unrelated to COVID-19 were not well defined in the EMR, the composite disease outcomes in our analysis were defined as 'all-cause' outcomes. Therefore, our findings may overestimate the actual COVID-19 related outcomes. Despite the limitations, our study is one of few studies reported the important longitudinal health consequences in young COVID-19 patients, at both inpatient and emergency department encounters.

## Conclusion

A significant portion of COVID-19 patients 18–29 years old in this cohort experienced serious disease outcomes, demonstrating the risk of severe disease even among young adult populations and especially among members of marginalized communities and people living with obesity, asthma, cardiovascular disease, or diabetes. Additionally, a high proportion of patients returned to the hospital within thirty days of their initial diagnostic encounter, emphasizing the need for greater support for young adults diagnosed with COVID-19. Health authorities must emphasize COVID-19 awareness and prevention in young adults and continue investigating risk factors for severe disease, readmission and long-term sequalae in this population.

## Supporting information

**S1 Table. Characteristics of COVID patients aged 18–29 years, by time of severe disease or pneumonia diagnosis.**
(DOCX)

## Acknowledgments

The authors thank the Houston Methodist Center for Outcome Research's CURATOR team for providing data using in this analysis.

## Author Contributions

**Conceptualization:** Micaela Sandoval, Duc T. Nguyen, Edward A. Graviss.

**Data curation:** Micaela Sandoval, Duc T. Nguyen, Farhaan S. Vahidy, Edward A. Graviss.

**Formal analysis:** Micaela Sandoval.

**Supervision:** Edward A. Graviss.

**Writing – original draft:** Micaela Sandoval.

**Writing – review & editing:** Micaela Sandoval, Duc T. Nguyen, Farhaan S. Vahidy, Edward A. Graviss.

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
