## [Decision Letter · Decision Letter 0]

1 Jul 2021

PONE-D-21-13036

Risk Factors for Severity of COVID-19 in Hospital Patients Age 18-29 Years

PLOS ONE

Dear Dr. Graviss,

Thank you for submitting your manuscript to PLOS ONE. After careful consideration, we feel that it has merit but does not fully meet PLOS ONE’s publication criteria as it currently stands. Therefore, we invite you to submit a revised version of the manuscript that addresses the points raised during the review process.

We look forward to receiving your revised manuscript.

Kind regards,

Aleksandar R. Zivkovic

Academic Editor

PLOS ONE

3. We note that Figure 3 in your submission contain map images which may be copyrighted. All PLOS content is published under the Creative Commons Attribution License (CC BY 4.0), which means that the manuscript, images, and Supporting Information files will be freely available online, and any third party is permitted to access, download, copy, distribute, and use these materials in any way, even commercially, with proper attribution. For these reasons, we cannot publish previously copyrighted maps or satellite images created using proprietary data, such as Google software (Google Maps, Street View, and Earth). For more information, see our copyright guidelines: http://journals.plos.org/plosone/s/licenses-and-copyright.

3.1.    You may seek permission from the original copyright holder of Figure 3 to publish the content specifically under the CC BY 4.0 license. 

3.2.    If you are unable to obtain permission from the original copyright holder to publish these figures under the CC BY 4.0 license or if the copyright holder’s requirements are incompatible with the CC BY 4.0 license, please either i) remove the figure or ii) supply a replacement figure that complies with the CC BY 4.0 license. Please check copyright information on all replacement figures and update the figure caption with source information. If applicable, please specify in the figure caption text when a figure is similar but not identical to the original image and is therefore for illustrative purposes only.

Reviewers' comments:

Reviewer #1: Sandoval et al describe risk factors associated with severe COVID-19 in patients aged 18-29 years seen at a large hospital system in Texas.

General comments:

- Females who are pregnant during the pandemic visit the hospital because of regular check-ups of their pregnancy or labor. These hospital visits are unrelated to COVID-19 however these women were regularly screened for COVID-19 to prevent the spread of the disease. This group of COVID-19 positive patients is a particular group of COVID-19 patients, which may need to be highlighted more in the discussion. I am not sure if I agree with the author’s statement that pregnant women provide insight into disease dynamics of the general population. This only female group has different healthcare utilization needs than the general population.

- Throughout the abstract and manuscript, the authors report the OR / aOR with the p-value in the text. I have a preference for reporting the 95% confidence interval because the range of the confidence interval provides additional information over the p-value.

- The authors were only able to catch diagnosis/ readmissions within their 30-day time frame if patients presented again to one of the hospitals in the health system. The authors need to acknowledge that it may be possible that patients presented elsewhere which affects their outcomes.

- How did the authors decide on whether these diagnoses were COVID-19 related and therefore COVID-19 severe disease? It could of course very well be that a patient had a myocardial infarction unrelated to COVID-19 in the 30 days following a COVID-19 diagnosis?

Abstract:

- The sentence: “This study was limited to young adults diagnosed at a hospital encounter and results may not be generalizable to all COVID-19 patients” can be removed from the abstract as it is clear from the research question that this study focused solely on young adults.

Introduction:

- The authors mention that few (two) studies included a young patient population. Please include their findings in the introduction and explain why current study is different. This information may be more elaborately included in the discussion.

Methods

- I suggest to move the paragraph regarding geographic data collection and analyses to the end of the methods (just before the role of the funding source.

- How did the authors finalize the list of diagnoses that defined severe disease?

Results:

- Table 1: Please explain the area deprivation index in the methods; what does the scale 1 through 10 represent?

- Given that this a young study population and you would expect patients to be relatively healthy, would it not be better to have the cut-off values for the Charlson comorbidity index be: 0, 1, 2, > 2 or even 0, 1, >1 as most patients (77.6%) do not have a comorbidity? (Table 1)

- In the text, the authors mention that 43% of the patients reported cough, sore throat and/ or shortness of breath, while table 2 refers to these symptoms as respiratory symptoms. Can the authors describe which symptoms were included in each group of symptoms mentioned in table 2?

- What is the relevance of being hospitalized to a satellite hospital compared to the flagship hospital?

- Can the authors explain why “missing BMI” is relevant to describe as a protective factor for readmission?

- Can the authors explain the following inconsistencies in tables 4 through 6:

o Gender has a reference group in the univariable analysis and not in the multivariable analysis

o What is the meaning of the ** for certain covariates? Why are there no results for certain categories (for instance for: NH Asian, Satellite Hospital #3, certain months, any known exposure.. etc.) Seen the relatively low number of patients per covariate group, the authors should consider regrouping the covariates.

o I assume that the authors calculated the OR and aOR separately per medical history and surgical history. Can the authors explain why only the OR is reported for cholecystectomy (which is mentioned be the authors when describing table 1), while the aOR is reported for solid organ transplant (with a wide 95% confidence interval).

Discussion:

- The following sentence is contradictory and the word likewise does not seem to be correct as non-Hispanic Black patients were not more likely to be diagnosed with severe disease or pneumonia while Hispanic patients were, please revise: “Notably, non-Hispanic Black patients were not more likely to be diagnosed with severe disease or pneumonia, but they were likely to return to the hospital within thirty days, compared to White patients; likewise, Hispanic patients were more likely to be diagnosed with severe disease or pneumonia compared to White patients”

- Given the amount of information provided by the authors in the results, I find the discussion relatively short and lacking details/ in depth descriptions of the findings and comparison with already published manuscripts. Topics of interest may include: pregnancy, results of other studies that included younger patient populations, information of marginalized populations (as this is highlighted in the abstract), etc.

References:

- Please update references when more details about recently published manuscripts are currently available (e.g. Cunningham JW, Vaduganathan M, Claggett BL, Jering KS, Bhatt AS, Rosenthal N, et al. Clinical Outcomes in Young US Adults Hospitalized With COVID-19)

Reviewer #2: Abstract/Methods: Please explain the acronym CHF

Introduction: 1. 'were young adults, aged 18-29 years,.' : no coma before full-stop.

Methods: Please explain , what do you mean "Patients were included if they received a positive diagnostic result from a severe acute respiratory syndrome coronavirus 2 (SARS-CoV-2) RNA polymerase chain reaction (PCR) assay".is there a PCR assay for a severe acute respiratory syndrome SARS-CoV-2?

Methods/ Electronic Medical Record Data Collection:The authors state that" BMI was calculated and classified according to the Centers for Disease Control and Prevention (CDC) and World guidelines Health Organization (WHO)" You provide the same reference for CDC and WHO as well.

Reviewer #3: To the authors

The current manuscript is interesting and important. The authors found some factors associated with severity in young COVID-19 patients. The current manuscript involves several factors associated with short- and long-term health consequences of this emerging infectious disease in young adults

My questions:

Methods

+Acute kidney injury is associated with poor prognosis in COVID-19 patients. I have not found how many patients had AKI. Do you have any data about AKI in those patients?

+I have found no reports of the use of psychoactive drugs. Do you have any record about psychiatric disease and drug users? Is there any relation between severe COVID-19 and both psychiatric disease and drug users?

Discussion

+In my opinion, you should write about the therapies used, such as azithromycin and dexamethasone, used in these patients in the discussion.

Reviewer #4: Sandoval et al. conducted a retrospective observational study of n=1853 young adults (18–29 years) seen in the ED or admitted to hospital at the time of a positive respiratory swab for SARS-CoV-2 in a single healthcare system in Houston, TX. Using logistic regression models, they evaluated risk factors for pneumonia, “other severe disease outcomes,” and hospital admission or readmission within 30 days. Thank you for the opportunity to review this manuscript.

1. Please provide more details about the initial hospital encounters. For example, what proportion of these were ED visits only? What proportion of individuals were initially hospitalized? Did the associations reported differ across these strata?

2. Some terminology used here is difficult. To call the original hospital encounter an “admission” may be unclear to some readers. I would suggest referring to this as a “hospital encounter.”

3. In addition, the term “readmission” could include a hospital readmission for individuals who were originally admitted or a first admission for individuals who were seen in the ED. I think it would be more clear to call this outcome a “second hospital encounter” and not a readmission.

4. Similarly, the outcome described as “severe disease” is also confusing. In COVID-19, the CDC defines severe outcomes as hospitalization, ICU admission, intubation, and death, but here the authors have something different in mind. This composite outcome should be described in a different way. Alternatively, consider using a more standard endpoint, such as ICU admission or mechanical ventilation.

5. Why were pregnant individuals excluded from some analyses?

6. Please include the covariates that were considered in the variable selection step. How were variables discarded?

Minor comments

The authors write, “(few previous studies) have incorporated longitudinal clinical data.” For clarity, I’d suggested rephrasing this to state, “Among studies in young adults, few (or none?) have incorporated longitudinal clinical data.”

6. PLOS authors have the option to publish the peer review history of their article (what does this mean?). If published, this will include your full peer review and any attached files.

Reviewer #1: No

Reviewer #2: **Yes: **Sophia Tsabouri, Associate Professor of Paediatrics & Paediatric Allergy

Faculty of Medicine, School of Health Sciences

University of Ioannina

Reviewer #3: No

Reviewer #4: No

---

## [Author Response · Author response to Decision Letter 0]

14 Jul 2021

Please find attached the "Response to Reviewers" file

---

## [Editor Report · Decision Letter 1]

19 Jul 2021

Risk Factors for Severity of COVID-19 in Hospital Patients Age 18-29 Years

PONE-D-21-13036R1

Dear Dr. Graviss,

We’re pleased to inform you that your manuscript has been judged scientifically suitable for publication and will be formally accepted for publication once it meets all outstanding technical requirements.

Kind regards,

Aleksandar R. Zivkovic

Academic Editor

PLOS ONE

---

## [Editor Report · Acceptance letter]

23 Jul 2021

PONE-D-21-13036R1 

Risk Factors for Severity of COVID-19 in Hospital Patients Age 18-29 Years 

Dear Dr. Graviss:

I'm pleased to inform you that your manuscript has been deemed suitable for publication in PLOS ONE. Congratulations! Your manuscript is now with our production department. 

Kind regards, 

on behalf of

Dr. Aleksandar R. Zivkovic 

Academic Editor

PLOS ONE